# Study of Biomass Composite Workpiece Support Structure Based on Selective Laser Sintering Technology

**DOI:** 10.3390/ma16134644

**Published:** 2023-06-27

**Authors:** Tianai Sun, Yanling Guo, Jian Li, Yifan Guo, Xinyue Zhang, Yangwei Wang

**Affiliations:** College of Mechanical and Electrical Engineering, Northeast Forestry University, Harbin 150040, China; sta0512@nefu.edu.cn (T.S.); pariss.guo@nefu.edu.cn (Y.G.); 2021125909@nefu.edu.cn (X.Z.); wang.yangwei@nefu.edu.cn (Y.W.)

**Keywords:** selective laser sintering, 3D printing, thin-walled parts, warpage deformation, warpage suppression, support structures

## Abstract

When using selective laser sintering to print parts with thin-walled structures, the thermal action of the laser can cause thermal stresses that lead to plastic deformation, resulting in large warpage and dimensional deviations. To address this issue, this study proposes a bottom support method for selective laser sintering. The impact of lattice-type, concentric-type, and cross-type support structures with varying filling densities and thicknesses on the suppression of warpage and dimensional errors was investigated. The optimal process parameters for each support structure were then determined through optimization. The findings of this study demonstrated a reduction in *Z*-axis dimensional errors of the workpiece following the addition of supports. The reduction amounted to 33.809%, 86.160%, and 66.214%, respectively, compared to the original workpiece. Moreover, the corresponding warpage was reduced by 35.673%, 46.189%, and 46.059% for each respective case, showcasing an improvement in the printing precision. Therefore, the bottom support effectively reduces dimensional and shape errors in thin-walled parts printed by selective laser sintering. Specifically, the results obtained indicated that the concentric type of support is more effective in reducing dimensional errors and enhancing the shape accuracy of the printed workpiece. Conversely, the cross type of support demonstrated superior capabilities in minimizing the consumption of printing materials while still delivering satisfactory results. Thus, this study holds promise for contributing to the advancement of thin-walled part quality using selective laser sintering technology. This research can contribute to achieving greater accuracy in the fabrication of parts through 3D printing.

## 1. Introduction

Selective laser sintering is an additive manufacturing technique renowned for its high efficiency, low cost, and superior quality [1]. Consequently, it has gained significant traction in diverse industrial sectors, including aviation and automotive manufacturing [2,3]. In selective laser sintering, a laser is deployed as a heat source to melt powder material along a predetermined path for bonding purposes [4,5,6]. However, due to the process’s susceptibility to thermal and residual stresses, warpage deformation is common, leading to significant dimensional errors, especially in thin-walled parts [7,8,9]. Consequently, research efforts have focused on developing effective methodologies to counter residual thermal stresses and suppress corresponding warpage deformation.

The suppression of warpage deformation during the selective laser sintering process is a crucial factor for improving workpiece forming accuracy and has been extensively studied in the field of rapid prototyping technology [10,11]. Zhang et al. [12] found that the degree of warpage deformation of the workpiece under different scanning strategies showed significant differences and investigated the effects of varying scanning paths on workpiece warpage and surface morphology through experimental means. Gouveia et al. [13] proposed the impact of support structure and laser patterning strategy on part surface quality and forming accuracy during heat treatment, and experiments indicated that a combination of support structure and streak 67° strategy produced the lowest residual stress. Gulcan et al. [14] explored the optimization of non-contact support parts against part deformation due to high thermal gradients. Performance indicators such as microhardness between thicknesses, microstructural changes on the bottom surface, surface roughness, and dimensional changes were used to experimentally verify the effects of non-contact support thickness and two support spacings. It was emphasized that the optimal design of the support structure is a key direction for solving this problem. Huang et al. [15] proposed that the support structure takes away heat and reduces geometric deformation during the process. In addition, the effect of different support structures on the warpage deformation of the suspended beam part was studied. Gan et al. [16] conducted experiments and finite element analysis to investigate the effect of three types of support structures on the design of thin rectangular plates. The analysis concluded that the spacing of the support structures impacts the thermal deformation of the workpiece significantly, highlighting the optimal design of the structure as an effective way to reduce thermal stresses and suppress warpage deformation [17]. These studies are crucial for optimizing workpiece accuracy by mitigating workpiece warpage through support structures [18]. However, while these studies focused primarily on investigating the causes of workpiece warpage and the effects of support structures on thermal deformation, the specific degree of warpage deformation caused by variations in support structure thickness, density, and pattern remains inadequately quantified. Therefore, the quantitative relationship between workpiece warpage deformation and support structure parameters is presently uncertain, and further research is necessary to reveal this relationship.

According to the existing literature, some researchers are still using empirical or trial-and-error methods to investigate the influence of thickness, density, and pattern of the support structure on the degree of warpage deformation, which consumes more human and material resources to match the appropriate support structure and process parameters. The relationship between the thickness, density, and pattern of the support structure and the degree of warpage deformation is not clearly quantified. In this paper, experiments will be designed to explicitly quantify the effects of support thickness, density, and pattern in the support structure on the warpage deformation of selective laser sintered workpieces. In this paper, experiments will be designed to explicitly quantify the effects of support thickness, density, and pattern in the support structure on the warpage deformation of selective laser sintered workpieces. The experimental results will also be analyzed. In addition, based on a large amount of experimental data, a prediction model of the three parameters of the support structure on the warpage deformation of the workpiece will be established to obtain the optimal combination of parameters of the support structure to reduce the warpage deformation of the workpiece. Eventually, it will reach the goal that warpage can be accurately predicted by the support thickness, density, and pattern in the support structure, and the optimal combination of support structure parameters to reduce warpage of the workpiece will be obtained. This will help improve the accuracy of the workpiece through selective laser sintering.

## 2. Materials and Methods

### 2.1. Materials

In this study, we used biomass composites, which consisted of Co-PES powder and walnut shell powder. Co-PES powder (particle diameter range of 0–60 µm, apparent density of 0.7 g/cm^3^) was copolyester hot melt adhesive powder provided by Anhui Tiannian Material Technology Ltd. (Chuzhou, China). Walnut shell powder (approximately spherical particles, particle diameter range of 58–100 µm, and apparent density of 0.48 g/cm^3^) was obtained from a food enterprise. Using a high-speed mixer with a heating function (SHR50A Hongji Machinery Co., Ltd., Zhangjiagang, China). The Co-PES powder and walnut shell powder were mixed at a ratio of 4:1 for 15 min under low-speed agitation below 30 °C, followed by an additional 5 min of high-speed mixing. The microscopic morphology of the biomass composites before and after mixing was observed using an optical microscope, as depicted in Figure 1.

Figure 1a depicts the microstructure of the walnut shell powder material under electron microscopy, revealing its irregular cluster formation. Figure 1c displays the walnut shell particles within the biomass composites are uniformly distributed in the Co-PES matrix material.

The selective laser sintering process is a highly intricate thermophysical-chemical procedure that utilizes a laser beam with high energy density to sinter and bond the workpieces stacked layer by layer as per a pre-designed scanning path. As such, the thermal properties of the material play a role in the subsequent forming process. To this end, the Yagi-Kunii model and the Kopp-Neulnann law, as proposed by Yu et al. [19], were deployed to determine the biomass composite’s density, specific heat capacity, and thermal conductivity of the powder.

Table 1 highlights the thermal and scientific features of the biomass composites based on experimental tests to derive the specific heat capacity and thermal conductivity. With increasing temperature, the nonlinearity of temperature dependences is evident. The values of the specific heat capacity and thermal conductivity are not proportional and linear, showing irregular movements and sudden changes. Additionally, the first-order derivatives are not constant. Thus, it can be inferred that, unlike homogeneous materials, Its specific heat capacity and thermal conductivity exhibit temperature-dependent and non-linear trends.

### 2.2. Analysis of Warpage Causes

The CO_2_ laser utilized in selective laser sintering generates a Gaussian heat source, with the most substantial energy output at the center of the laser spot and subsequently decreasing in intensity outward, as illustrated in Figure 2a. To showcase the temperature, rise process on the surface layer of the powder more effectively, a heat transfer finite element model was developed to simulate the effect of the laser on the powder temperature. At *t* = 0, the biomass composite powder is in the preheating stage, which usually involves the preheating temperature of the powder bed, and its heat transfer is expressed by Equation (1).
(1)$${\left.T\right|}_{t=0}=T_{w}$$

Here, *T_w_* refers to the preheating temperature, expressed in degrees Celsius (°C).

The heat transfer boundary conditions are depicted in Figure 2b. The boundary region ζ is characterized by the temperature distribution function *T*(*x,y,z,t*), which is subject to the first type of boundary conditions, as defined in Equation (2).
(2)$${\left.T\right|}_{{}_{\zeta }}=T\left(x,y,z,t\right)$$

The heat flow density function *q*(*x,y,z,t*) is computed using Fourier’s law, given the established boundary conditions. Specifically, the powder surface layer unaffected by the laser conforms to the second type of boundary conditions, outlined in Equation (3).
(3)$${\left.-\lambda \frac{𝜕T}{𝜕n}\right|}_{\zeta }=q\left(x,y,z,t\right)$$
where *λ* denotes the material’s thermal conductivity coefficient, measured in W/(m∙°C).

The radiation and convection heat transfer taking place between air and powder constitute the third category of boundary conditions, as illustrated in Equation (4).
(4)$${\left.-k\frac{𝜕T}{𝜕z}\right|}_{z=0}+h\left(T_{b}-T_{f}\right)+\sigma_{0}\varepsilon \left(T-T_{f}^{4}\right)=q$$
where *T_b_* denotes the surface temperature specified in degrees Celsius (°C); *T_f_* represents the ambient temperature of the surrounding space in degrees Celsius (°C); *h* denotes the convective heat coefficient between the equivalent unit and the air, expressed in W/(m^2^∙°C); *σ*_0_ represents the Stefan-Boltzmann constant, with a value of approximately 5.67 × 10^−8^ W/(m^2^∙K^4^); *ε* denotes the thermal radiation coefficient, while *q* corresponds to the heat flow density upon heating in W/m^2^.

The findings from the heat transfer simulation of the powder surface layer are presented in Figure 2c. The outcome suggests that the surface layer charges heat faster than the bottom layer, leading to greater volume shrinkage in the fabricated part. This, in turn, generates uneven stress distribution, mainly in the form of shear and tangential stresses, causing warpage of the part, as depicted in Figure 2d.

### 2.3. Methods

Jonkers et al. [20] reported that support structures aid in retaining the shape of the workpiece, reducing the degree of warpage, and solving thermal deformation issues. To investigate how support thickness, density, and pattern affect the warpage of a workpiece in selective laser sintering support structures, three support patterns –grid type, concentric type, and cross type—were determined through extensive literature review and pre-testing. A two-factor, five-level CCD test was devised for each of the three support structures, totaling nine tests. Table 2 illustrates the factors and levels considered in the study. The measured test indexes included *Z*-axis dimensional error *E*, warpage amount *γ*, support printing time *t*, and consumable weight *m*.

Parts manufactured using rapid prototyping equipment (CX-B200, Liberty Smart Co., Harbin, China) were printed. The employed process parameters consisted of a preheating temperature of 80 °C, with a maximum temperature error not exceeding 5 °C. Laser power was set to 12 W, scanning speed at 3000 mm/s, layer thickness at 0.1 mm, and scanning spacing at 0.1 mm. As illustrated in Figure 3, the biomass composite powder was evenly laid flat in the rapid prototyping machine’s powder supply and forming boxes, and the designed part size was 80 mm × 16 mm × 2 mm for testing. Upon reaching the preheating temperature, processing was initiated. After maintaining the workpiece at a temperature of 50 °C for a duration of 30 min following the completion of machining, the support was removed, and the Z-directional dimension was meticulously measured to calculate the extent of warpage. Additionally, the printing time of the support and the weight of the consumed materials were recorded.

Equation (5), as illustrated in Figure 4, calculates the warpage amount.
(5)$$\gamma =\frac{h}{L}$$

*γ*, representing the warpage amount in mm, is calculated using Equation (5), shown in Figure 4. Here, the variable *h* represents the maximum distance in mm the workpiece and the horizontal direction, while *L* is the projected length of the workpiece in the horizontal direction, also in mm.

The weight of the consumables is calculated as shown in Equation (6).
(6)$$m=\rho \cdot V=\rho \cdot \pi R^{2}l$$
where: *m* is the weight of the consumable in g; *ρ* is the density of this biomass composite; *V* is the volume of the material required for printing; *R* is the laser spot radius in mm, and *l* is the laser alignment distance in mm.

Through the quantification of the workpiece’s warpage degree, a regression model can be employed to determine the level of warpage considering different factors. Subsequently, a multi-objective optimization approach that integrates support costs can be utilized to solve the optimal support structure.

## 3. Results and Discussions

The experiment was conducted at the 3D printing center of Northeast Forestry University. The room temperature during the test was 25 °C. The workpieces with different support structures are shown in Figure 5.

### 3.1. Analysis of Experimental Results of Grid-Type Supports

The test outcomes of the grid-type support structure are presented in Table 3. To reduce any potential test errors, six workpieces were printed simultaneously, and the results obtained were calculated by taking an average of the measured data from the six workpieces.

Table 4 displays the outcomes of the ANOVA pertaining to the obtained test results. In the table, *p* > 0.05 suggests the model’s insignificance, and 0.01 ≤ *p* ≤ 0.05 demonstrates the model’s significance. Lastly, *p* < 0.001 indicates the model’s high level of significance. From the analysis of Table 4, it is evident that the regression model of the *Z*-axis dimensional error *E* is significant, while the models for warpage *γ*, support printing time *t*, and consumable weight *m* are all highly significant.

Equation (7) indicates the regression polynomial for the *Z*-axis dimensional error *E* of the biomass composite workpiece. The model possesses an R^2^ = 0.8438, while the Adjusted R^2^ = 0.75. Further, Predicted R^2^ = 0.5708, and the signal-to-noise ratio of this polynomial is 8.3901, which is greater than 4, thus indicating the model’s applicability.
(7)$$E=-0.127515\delta -0.001008\omega +0.051333\delta^{2}+0.365343$$

The regression polynomial for the warpage amount *γ* of the biomass composite workpiece is shown in Equation (8). The model has R^2^ = 0.8647, Adjusted R^2^ = 0.8196, Predicted R^2^ = 0.7436, and the signal-to-noise ratio of this polynomial is 10.0058 > 4. This indicates that the model is usable.
(8)γ=−1.05088δ−0.0232266ω+2.71780

The regression polynomial for the biomass composite workpiece support printing time *t* is shown in Equation (9). The model has R^2^ = 0.9995, Adjusted R^2^ = 0.9992, Predicted R^2^ = 0.9986, and the signal-to-noise ratio of this polynomial is 145.1483 > 4. This indicates that the model is usable.
(9)t=2.46751δ−0.082690ω+0.55δω−1.14704

The regression polynomial for the weight of biomass composite workpiece consumables *m* is shown in Equation (10) with R^2^ = 0.9988, Adjusted R^2^ = 0.9981, Predicted R^2^ = 0.9957, and the signal-to-noise ratio of this polynomial is 96.7241 > 4. This indicates that the model is usable.
(10)m=0.33864δ−0.021998ω+0.095165δω+17.5889

The response surfaces between each indicator and the factors are shown in Figure 6. Figure 6a indicates that as the support density *ω* increases, there is a decline in the *Z*-axis dimensional error *E*, and an evident inverse relationship exists between them. This finding is attributable to the grid-type support pattern consisting of small columnar structures uniformly distributed on the workpiece’s surface, which run parallel to the workpiece and possess a small support area. If the grid-type support structure’s density is insufficient, the support structure will experience vibration and deformation, causing the *Z*-axis dimensional error. It is further apparent from Figure 6a that the *Z*-axis dimensional error initially decreases and then increases with the support thickness *δ* increase, as the first sintering layer accumulates more energy under unchanged conditions. A larger support thickness decreases the impact of the initial few layers’ energy accumulation on the workpiece’s *Z*-axis dimensional accuracy. Still, the *Z*-axis dimensional error decreases further with a rise in the support thickness due to the continuous accumulation of energy in the bottom layer during the sintering process. As the sintering depth grows, the energy is absorbed by each layer, leading to an energy difference between the upper and lower surfaces of the powder, resulting in deformation due to excess energy in the bottom layer. Consequently, after a particular thickness is attained, the *Z*-axis dimensional error significantly increases.

Figure 6b depicts that the warpage *γ* notably decreases with the increase in support density *ω* and thickness *δ*. This phenomenon is due to the presence of inhomogeneous thermal stress and shrinkage causes warping. However, as the density and thickness of the support structure increase, laser energy gradually accumulates, and energy constraints and compensation between each layer occur. Therefore, under the same process parameters, the warping phenomenon diminishes slowly.

Figure 6c,d highlight that the support printing time *t* and consumable weight *m* exhibit an increasing tendency as both support density *ω* and support thickness *δ* increase. Thus, it is vital to explore the optimal solution to reduce the *Z*-axis dimensional error and warpage while aiming to minimize printing time *t* and consumable weight *m*.

### 3.2. Analysis of Test Results of Concentric Type Support

The test outcomes of the concentric type of support structure are presented in Table 5. To reduce any potential test errors, six workpieces were printed simultaneously, and the results obtained were calculated by taking an average of the measured data from the six workpieces.

The results of the ANOVA on the test results are shown in Table 6. From Table 6, it can be seen that the regression models of warpage amount *γ* and *Z*-axis dimensional error *E* are significant. The models for support printing time *t* and consumable weight *m* are both highly significant.

The regression polynomial for the *Z*-axis dimensional error *E* of the biomass composite workpiece is shown in Equation (11). The model has R^2^ = 0.8463, Adjusted R^2^ = 0.6708, Predicted R^2^ = 0.6708, and the signal-to-noise ratio of this polynomial is 11.2462 > 4. This indicates that the model is usable.
(11)$$E=-0.276519\delta +0.002918\omega +0.17789\delta^{2}+0.18375$$

The regression polynomial for the warpage amount *γ* of the biomass composite workpiece is shown in Equation (12). The model has R^2^ = 0.776, Adjusted R^2^ = 0.7014, Predicted R^2^ = 0.5040, and the signal-to-noise ratio of this polynomial is 7.6956 > 4. This indicates that the model is usable.
(12)γ=−0.263034δ+0.008223ω+1.03024

The regression polynomial for the biomass composite workpiece support printing time *t* is shown in Equation (13). The model has R^2^ = 0.9997, Adjusted R^2^ = 0.9994, Predicted R^2^ = 0.9984, and the signal-to-noise ratio of this polynomial is 174.7081 > 4. This indicates that the model is usable.
(13)t=1.03553δ−0.008579ω+0.3δω−0.863961

The regression polynomial for the weight of the biomass composite workpiece consumables m is shown in Equation (14). The model has R^2^ = 0.9998, Adjusted R^2^ = 0.9998, Predicted R^2^ = 0.9996, and the signal-to-noise ratio of this polynomial is 272.6389 > 4. This indicates that the model is usable.
(14)m=0.072042δ−0.025382ω+0.098872δω+17.69248

The response surfaces of each indicator and factor are shown in Figure 7. As evident from Figure 7a, there exists a positive relationship between the *Z*-axis dimensional error *E* and the support density *ω*, with an increase in one resulting in an increase in the other. This finding is due to the high density of the concentric support pattern, which impedes heat transfer and results in high local thermal stress. In turn, this increases the impact of positive stress, causing *Z*-axis dimensional errors. Further, Figure 7a demonstrates that the *Z*-axis dimensional error has a slight decrease initially before increasing substantially with an increase in support thickness *δ*. This trend is due to the concentric ring structure of the concentric-type support pattern, which offers an average support area and has a limited impact on the workpiece’s *Z*-axis dimensional error. However, as support thickness further increases, deformation occurs via the sintering process, leading to continuous bottom layer energyaccumulation, causing large amounts of energy in the bottom layer powder. Consequently, the *Z*-axis dimensional error increases substantially after a certain thickness is attained.

Figure 7b illustrates a consistent decrease in warpage amount *γ* with in-brace thickness *δ*. Conversely, an increase in brace density *ω* is shown to result in an increase in warpage amount *γ*. These trends can be attributed to the uniform support spacing of the concentric-type support pattern. When the brace density *ω* increases, the support points come closer, and excessive contact between the support structures and the sintered part occurs. Excessive support force is applied to the workpiece, resulting in the generation of non-uniform thermal stress distribution within the workpiece and seriouspage. Excessive brace density causes the support points to be closer to each other, leading to excessive interaction. This excessive interaction exerts too much force on the sintered part, resulting in thermal stress generation during workpiece heat release. This phenomenon easily induces a non-uniform thermal stress distribution within the workpiece, ultimately resulting in serious warpage.

Figure 7c,d demonstrate that an increase in support density *ω* and thickness *δ* results in a gradual increase in both support printing time *t* and consumable weight *m*. Therefore, it is critical to seek optimal solutions that can reduce the *Z*-axis dimension and warpage while minimizing printing time and consumable weight. This strategy ensures that a shorter time and reduced consumable weight can be attained simultaneously, in addition to minimizing unwanted effects such as *Z*-axis error and warpage amount.

### 3.3. Analysis of Experimental Results of Cross-Type Supports

The test outcomes of the cross-type support structure are presented in Table 7. To reduce any potential test errors, six workpieces were printed simultaneously, and the results obtained were calculated by taking an average of the measured data from the six workpieces.

The results of the ANOVA on the test results are shown in Table 8. From Table 8, it can be seen that the regression models for the *Z*-axis dimensional error *E*, warpage amount *γ*, support printing time *t* and consumable weight m are extremely significant.

The regression polynomial for the *Z*-axis dimensional error *E* of the biomass composite workpiece is shown in Equation (15). The model has R^2^ = 0.9186, Adjusted R^2^ = 0.8697, Predicted R^2^ = 0.7078, and the signal-to-noise ratio of this polynomial is 12.9104 > 4. This indicates that the model can better represent the relationship between the factors and the *Z*-axis dimensional error *E* of the biomass composite workpiece.
(15)$$E=-0.164286\delta +0.000835\omega +0.091358\delta^{2}+0.3097$$

The regression polynomial for the warpage amount *γ* of the biomass composite workpiece is shown in Equation (16). The model has R^2^ = 0.776, Adjusted R^2^ = 0.7014, Predicted R^2^ = 0.5040, and the signal-to-noise ratio of this polynomial is 7.6956 > 4. This indicates that the model can better represent the relationship between the factors and the warpage amount *γ* of the biomass composite workpiece.
(16)$$\gamma =0.031765\delta -0.00824\omega -0.0287\delta^{2}+0.869856$$

The regression polynomial for the biomass composite workpiece support printing time *t* is shown in Equation (17). It can be seen that there is an interaction in the composition of this polynomial. The model has R^2^ = 0.9997, Adjusted R^2^ = 0.9994, Predicted R^2^ = 0.9984, and the signal-to-noise ratio of this polynomial is 174.7081 > 4. This indicates that the model can better represent the relationship between the factors and the biomass composite workpiece support printing time *t*.
(17)t=−3.11091δ−0.215901ω+0.6δω+2.31782

The regression polynomial for the weight of the biomass composite workpiece consumables *m* is shown in Equation (18). The model has R^2^ = 0.9998, Adjusted R^2^ = 0.9998, Predicted R^2^ = 0.9996, and the signal-to-noise ratio of this polynomial is 272.6389 > 4. This indicates that the model can better represent the relationship between the factors and the weight of biomass composite workpiece consumables *m*.
(18)m=0.235366δ−0.019276ω+0.078274δω+17.59013

The response surfaces between each indicator and the factors are shown in Figure 8. From Figure 8a, it is evident that, for the cross-type support, increasing support density *ω* to a decrease in *Z*-axis dimensional error *E*, showing an inverse relationship similar to that observed under the grid-type support pattern. Figure 8a also shows that the *Z*-axis dimensional error follows a decreasing and then increasing trend with increasing support thickness *δ*, but the magnitude of change is relatively smaller than that observed under the grid-type support pattern. This trend arises from the cross-type support pattern’s positioning of the support points at the workpiece’s geometric center, which helps provide uniform support and interference with other parts, ultimately minimizing the *Z*-axis dimensional error of the workpiece.

Figure 8b shows that the warpage amount *γ* decreases with increasing support thickness *δ*, while increasing with increasing support density *ω*. This pattern arises from the cross-type support’s even distribution of support points on the workpiece surface. This phenomenon allows for mutual and cross points, with the support force transferred in multiple directions, and the shorter distance between support points leads to more uniform support compared to concentric support. Thus, increasing support density and thickness under the type support pattern can effectively reduce workpiece warpage.

Figure 8c,d depict an increase in the support printing time *t* and consumable weight m as support density *ω* and thickness *δ* increase. Therefore, an optimal solution that minimizes both the workpiece *Z*-axis dimensional error and warpage amount, while prioritizing shorter printing times and lower consumable weights, is crucially important. By incorporating this approach, it is possible to achieve more efficient 3D printing processes characterized by reduced inaccuracies, printing time, and overall printing costs.

### 3.4. Analysis of Optimal Support Structure

Following the acquisition of regression models for each support structure, a multi-objective optimization approach was utilized to optimize the support structure process parameters of selective laser sintering while minimizing both the *Z*-axis dimensional error *E*, warpage amount *γ*, support printing time *t*, and consumable weight *m*. The optimized theoretical and actual results using the optimal parameters are summarized in Table 9. Notably, the grid-type support structure demonstrated *z*-axis dimensional errors *E*, warpage amount *γ*, support printing time *t*, and consumable weight *m* errors of 4.06%, 1.78%, 0.01%, and 0.01%, respectively. The concentric type of support structure exhibited a *z*-axis dimensional error *E*, warpage amount *γ*, support printing time *t*, and consumable weight *m* errors of 2%, 2.02%, 0.03%, and 0.02%, respectively. In contrast, the cross-type support structure demonstrated a *z*-axis dimensional error *E*, warpage amount *γ*, support printing time *t*, and consumable weight *m* errors of 2.89%, 4.802%, 0.01%, and 0.01%, respectively. This error is the relative error derived from the ratio of the actual error to the predicted value. Notably, the errors between the predicted values of the optimal solutions of the model and the experimental results of various support patterns were less than 5%. This outcome denotes the reliability of the model and the optimization results.

Furthermore, Table 9 depicts that the concentric support displays a comparatively smaller *Z*-axis dimensional error, warpage, and printing time, albeit at the cost of a larger consumable weight in contrast to the crossed support pattern. Notably, the grid-type support does not confer any significant advantages when compared to the other two support structures in terms of *Z*-axis dimensional error, warpage, print time, and consumable weight. Specifically, the cross-type support has smaller warpage compared to the grid-type support, and its *Z*-axis dimensional error, warpage, and support printing time rank in the middle among the three support structures, though with a relatively lower actual consumable weight. The optimization effects are evident in Figure 9.

It is imperative to note that this study only optimized the process parameters for a pre-defined support pattern and did not quantitatively establish which support pattern is superior. Therefore, future studies could focus on exploring support topology to obtain optimal support structures characterized by reduced consumable weight and printing time while guaranteeing improved overall performance. By doing so, it is possible to obtain more effective support structures that are both time and cost-efficient.

## 4. Conclusions

In this investigation, the chosen experimental material was a biomass composite comprising Co-PES powder and walnut shell powder in a 4:1 ratio. By employing the Yagi-Kunii model and the Kopp-Neulnann law, the density, specific heat capacity, and thermal conductivity of the biomass composite powder were accurately obtained. The density of the biomass composite powder exhibited an increasing trend within a temperature range of 20 °C to 300 °C, whereby the density stabilized once the temperature exceeded 100 °C. Meanwhile, the specific heat capacity demonstrated an initial increasing trend followed by a decreasing trend. Furthermore, the thermal conductivity of the material demonstrated a general increasing trend. These observations serve as fundamental parameters for the warpage model of the biomass composite.

The warpage phenomenon in selective laser sintering is a multifaceted issue, with the root cause being attributed to plastic deformation resulting from non-uniform heating. However, optimizing process parameters can enhance the performance of support structures. Based on the evaluated *Z*-axis dimensional error, warpage amount, support printing time, and weight of consumables, optimal parameters for the grid-type, concentric-type, and cross-type support structures were obtained. Specifically, the optimal parameters for grid-type support were 7% support density and 1.5 mm support thickness, while the optimal parameters for concentric-type support were 6% support density and 1.0 mm support thickness. Finally, the optimal parameters for cross-type support were 8% support density and 1.3 mm support thickness.

After optimizing the process parameters in selective laser sintering, a substantial reduction in both *Z*-axis dimensional error and warpage of the workpiece was observed when the corresponding support structure was added. Specifically, the *Z*-axis dimensional errors of the corresponding support structures were reduced by 33.809%, 86.160%, and 66.214%, respectively, whereas the corresponding warpage underwent a decrease of 35.673%, 46.189%, and 46.059%. Notably, these values were compared to the original workpiece, which had initial *Z*-axis dimensional error and warpage values of 0.737 mm and 1.548 mm, respectively.

Ultimately, these findings establish that the use of support structures in selective laser sintering is an effective means of reducing warpage and improving the z-directional dimensional accuracy of thin-walled parts. In terms of optimizing the dimensional accuracy and shape precision, the concentric type of support appears to be relatively more effective, whereas the cross type of support seems to excel in reducing the weight of consumables required for printing. Future research could be focused on expanding this approach to other applications with varying complexities.

## Figures and Tables

**Figure 1 materials-16-04644-f001:**
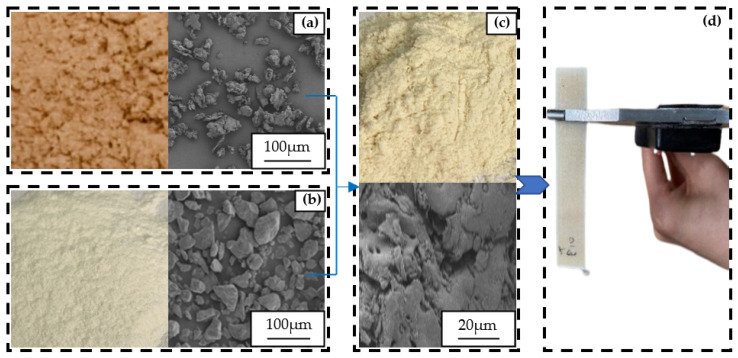
Microscopic morphology of biomass composites before and after mixing. (**a**) Walnut shell powder materials and their electron micrographs; (**b**) Co-PES powder materials and their electron micrographs; (**c**) Biomass composite powder materials and their electron micrographs; (**d**) Biomass composite workpieces.

**Figure 2 materials-16-04644-f002:**
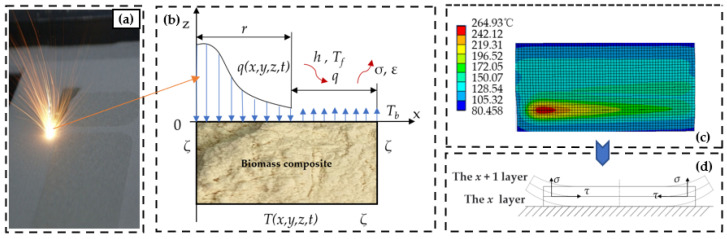
Analysis of the selective laser sintering warpage principle. (**a**) Selective laser sintering laser sintering process; (**b**) Heat transfer boundary conditions of the sintering process in the x-z direction cross-section; (**c**) Heat transfer simulation results of the powder surface layer; (**d**) Stress conditions on the warpage of the workpiece.

**Figure 3 materials-16-04644-f003:**
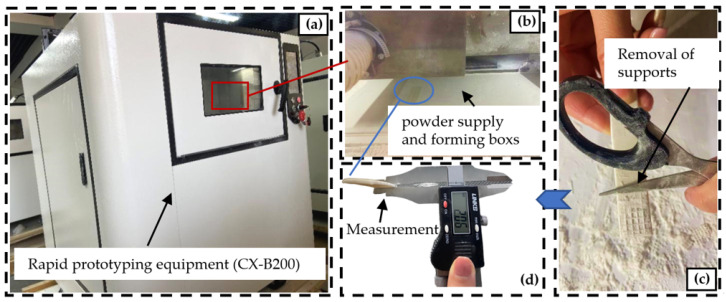
Test methods. (**a**) CX-B200 rapid prototyping equipment; (**b**) Selective laser sintering sintering process; (**c**) Support removal process; (**d**) Measurement of workpiece.

**Figure 4 materials-16-04644-f004:**
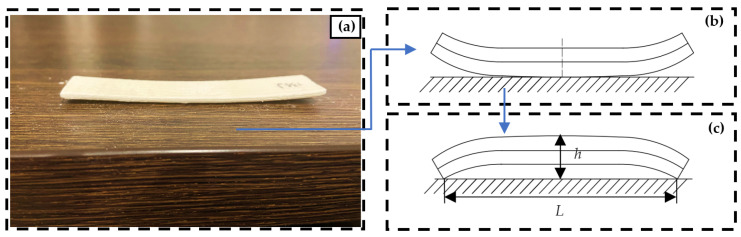
Selective laser sintering warpage calculation method. (**a**) warpage of the workpiece on the table; (**b**) the form of the workpiece on the table; (**c**) the diagram of the method of measuring the workpiece on the table after turning it over.

**Figure 5 materials-16-04644-f005:**
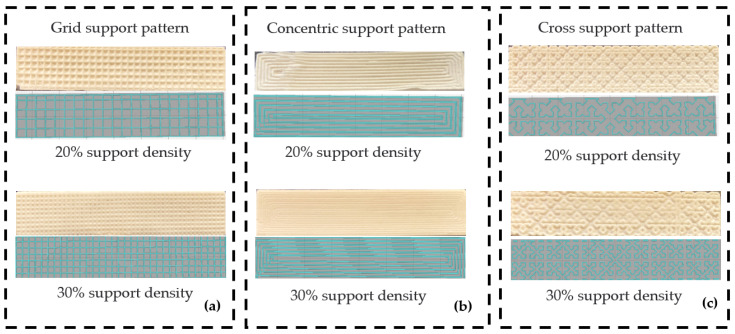
Comparison of expected and actual printing results for different support densities. (**a**) Comparison of expected and actual printing results for 20% and 30% support densities with grid-type support patterns; (**b**) Comparison of expected and actual printing results for 20% and 30% support densities with concentric support patterns; (**c**) Comparison of expected and actual printing results for 20% and 30% support densities with cross-type support patterns.

**Figure 6 materials-16-04644-f006:**
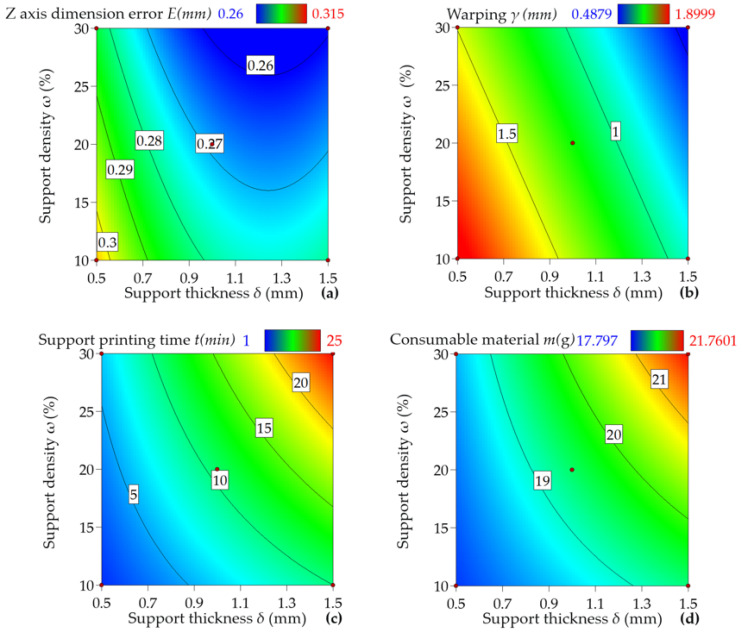
Data visualization of experimental results. (**a**) Effect of support density *ω* and thickness *δ* on the dimensional error *E* of *Z*-axis under the grid-type support pattern. (**b**) Effect of support density *ω* and thickness *δ* on warpage *γ* under grid-type support patterns. (**c**) Effect of support density *ω* and thickness *δ* on support printing time *t* under grid-type support patterns. (**d**) Effect of support density *ω* and support thickness *δ* on the weight of consumables *m* under the grid-type support pattern.

**Figure 7 materials-16-04644-f007:**
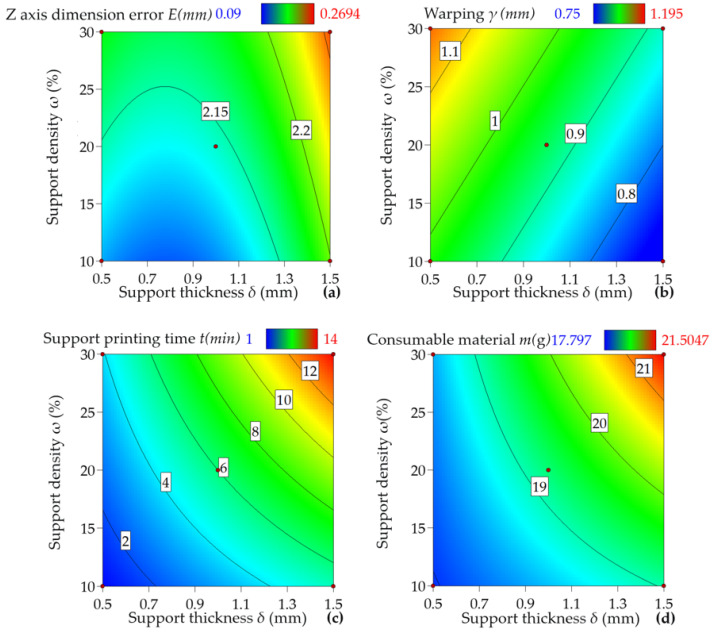
Data visualization of experimental results. (**a**) Effect of support density ω and thickness *δ* on *z*-axis dimensional error *E* under concentric support pattern; (**b**) Effect of support density *ω* and thickness *δ* on warpage amount *γ* under concentric support pattern; (**c**) Effect of support density *ω* and thickness *δ* on support printing time *t* under concentric support pattern; (**d**) Effect of support density *ω* and thickness *δ* on consumable weight *m* under concentric support pattern.

**Figure 8 materials-16-04644-f008:**
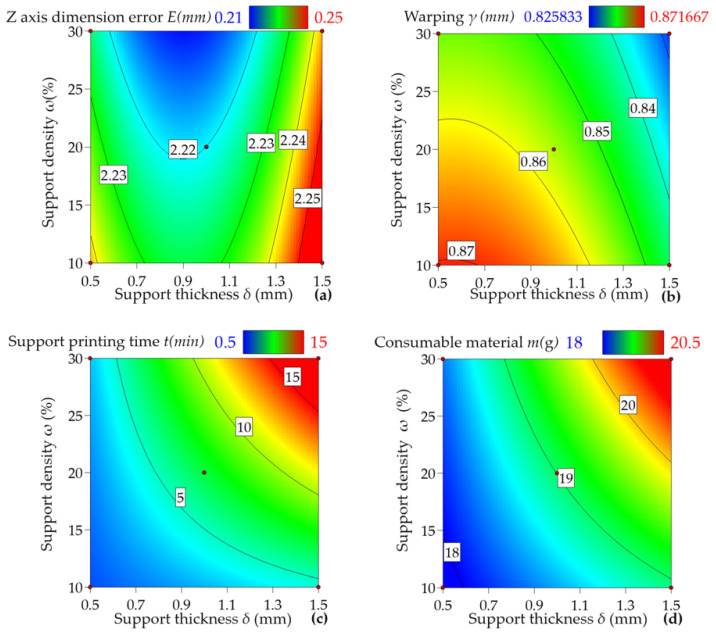
Data visualization of experimental results. (**a**) Effect of support density *ω* and thickness *δ* on *z*-axis dimensional error E under cross-type support patterns; (**b**) Effect of support density *ω* and thickness *δ* on warpage amount *γ* under cross-type support patterns; (**c**) The effects of support density *ω* and thickness *δ* on support printing time *t* under crossed support patterns; (**d**) The effects of support density *ω* and thickness *δ* on consumable weight *m* under crossed support patterns.

**Figure 9 materials-16-04644-f009:**
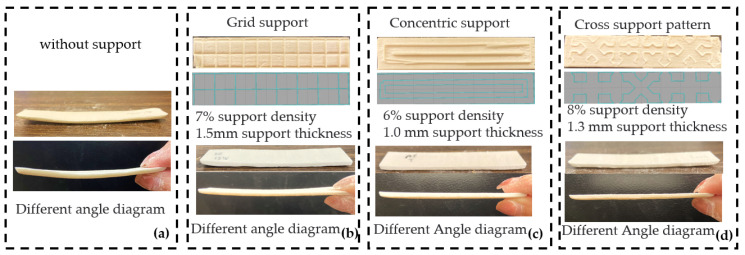
Optimal parameter test results. (**a**) The test results without the support; (**b**) The test results for the support density of 7% and the support thickness of 1.5 mm under the grid-type of support pattern; (**c**) The test results for the support density of 6% and the support thickness of 1.0 mm under the concentric type of support pattern; (**d**) The test results for the support density of 8% and the support thickness of 1.3 mm under the cross type of support pattern.

**Table 1 materials-16-04644-t001:** Biomass composite powder material thermal property parameters.

Temperature (°C)	Density (kg/m^3^)	Specific Heat Capacity (J/kg∙°C)	Thermal Conductivity (W/m∙°C)
20	656	2005	0.172
100	992	2105	0.243
160	992	2177	0.294
240	992	2244	0.375
300	992	2195	0.423

**Table 2 materials-16-04644-t002:** Factors and levels of CCD tests.

Level	Support Density *ω* (%)	Support Thickness *δ* (mm)
−1.682	5.85786	0.292893
−1	10	0.5
0	20	1
+1	30	1.5
+1.682	34.1421	1.70711

**Table 3 materials-16-04644-t003:** Test results of grid-type support.

No.	Factors		Indicators			
	*ω* (%)	*δ* (mm)	*E* (mm)	*γ* (mm)	*t* (min)	*m* (g)
1	20	0.292893	0.315	1.8999	1	17.79702
2	34.1421	1	0.26	0.71	17	20.43361
3	5.85786	1	0.281667	1.61833	4	18.45617
4	10	0.5	0.315	1.89	2	17.9618
5	20	1.70711	0.273	0.4879	20	20.92797
6	10	1.5	0.275	1.0667	10	19.2801
7	20	1	0.264	0.761667	11	19.3625
8	30	0.5	0.27	1.88167	6	18.53856
9	30	1.5	0.27	0.498333	25	21.76015

**Table 4 materials-16-04644-t004:** Analysis of the variance of test results under grid-type support.

Indicators	Source	Sum of Squares	df	Mean Square	F-Value	*p*-Value
** *E* **	Model	0.0029	3	0.001	9	0.0185
	*ω*	0.0012	1	0.0012	11.69	0.0189
	*δ*	0.0008	1	0.0008	7.69	0.0392
	*ω* ^2^	0.0008	1	0.0008	7.62	0.0398
	Residual	0.0005	5	0.0001		
	Cor Total	0.0034	8			
** *γ* **	Model	2.64	2	1.32	19.17	0.0025
	*ω*	2.21	1	2.21	32.06	0.0013
	*δ*	0.433	1	0.433	6.29	0.0461
	Residual	0.4133	6	0.0689		
	Cor Total	3.06	8			
** *t* **	Model	567.7	3	189.23	3159.97	<0.0001
	*ω*	362.75	1	362.75	6057.45	<0.0001
	*δ*	174.7	1	174.7	2917.32	<0.0001
	*ωδ*	30.25	1	30.25	505.14	<0.0001
	Residual	0.2994	5	0.0599		
	Cor Total	568	8			
** *m* **	Model	15.24	3	5.08	1379.08	<0.0001
	*ω*	10.05	1	10.05	2728.83	<0.0001
	*δ*	4.28	1	4.28	1162.57	<0.0001
	*ωδ*	0.9056	1	0.9056	245.84	<0.0001
	Residual	0.0184	5	0.0037		
	Cor Total	15.26	8			

**Table 5 materials-16-04644-t005:** Experimental results of various factors influenced by process parameters under a concentric support pattern.

No.	Factors		Indicators			
	*ω* (%)	*δ* (mm)	*E* (mm)	*γ* (mm)	*t* (min)	*m* (g)
1	20	0.292893	0.19333	1.18167	1	17.797
2	34.1421	1	0.18	1.195	10	20.2688
3	5.85786	1	0.09	0.83	2	18.209
4	10	0.5	0.09167	0.943333	1	17.9618
5	20	1.70711	0.2694	0.75	11	20.6808
6	10	1.5	0.2333	0.786667	5	19.0329
7	20	1	0.15833	0.826667	6	19.1977
8	30	0.5	0.18167	1.07833	4	18.4562
9	30	1.5	0.2495	0.793333	14	21.5047

**Table 6 materials-16-04644-t006:** Analysis of the variance of test results under concentric type support.

Indicators	Source	Sum of Squares	df	Mean Square	F-Value	*p*-Value
** *E* **	Model	0.029	3	0.0097	15.69	0.0056
	*ω*	0.0126	1	0.0126	20.36	0.0063
	*δ*	0.0068	1	0.0068	11.04	0.0209
	*ω* ^2^	0.0097	1	0.0097	15.67	0.0108
	Residual	0.0031	5	0.0006		
	Cor Total	0.0321	8			
** *γ* **	Model	0.1925	2	0.0962	10.4	0.0112
	*ω*	0.1384	1	0.1384	14.95	0.0083
	*δ*	0.0541	1	0.0541	5.84	0.0521
	Residual	0.0555	6	0.0093		
	Cor Total	0.248	8			
** *t* **	Model	175.94	3	58.65	4775.76	<0.0001
	*ω*	99	1	99	8061.71	<0.0001
	*δ*	67.94	1	67.94	5532.68	<0.0001
	*ωδ*	9	1	9	732.9	<0.0001
	Residual	0.0614	5	0.0123		
	Cor Total	176	8			
** *m* **	Model	13.7	3	4.57	11000.28	<0.0001
	*ω*	8.4	1	8.4	20237.46	<0.0001
	*δ*	4.32	1	4.32	10408.42	<0.0001
	*ωδ*	0.9776	1	0.9776	2354.96	<0.0001
	Residual	0.0021	5	0.0004		
	Cor Total	13.7	8			

**Table 7 materials-16-04644-t007:** Experimental results of various factors influenced by process parameters under cross-type support patterns.

No.	Factors		Indicators			
	*ω* (%)	*δ* (mm)	*E* (mm)	*γ* (mm)	*t* (min)	*m* (g)
1	20	0.292893	0.24767	0.863333	1	17.797
2	34.1421	1	0.20278	0.845833	11	19.7745
3	5.85786	1	0.2316	0.87015	2	18.1266
4	10	0.5	0.2467	0.871667	1	17.8794
5	20	1.70711	0.2713	0.825833	12	20.2688
6	10	1.5	0.26467	0.837667	5	18.9505
7	20	1	0.2133	0.858333	6	19.0329
8	30	0.5	0.2315	0.84667	4	18.2914
9	30	1.5	0.25383	0.831167	20	20.928

**Table 8 materials-16-04644-t008:** Analysis of the variance of test results under cross-type support.

Indicators	Source	Sum of Squares	df	Mean Square	F-Value	*p*-Value
** *E* **	Model	0.0038	3	0.0013	18.81	0.0037
	*ω*	0.0007	1	0.0007	10.12	0.0245
	*δ*	0.0006	1	0.0006	8.31	0.0345
	*ω* ^2^	0.0026	1	0.0026	37.99	0.0016
	Residual	0.0003	5	0.0001		
	Cor Total	0.0041	8			
** *γ* **	Model	0.0021	3	0.0007	26.34	0.0017
	*ω*	0.0013	1	0.0013	49.25	0.0009
	*δ*	0.0005	1	0.0005	20.34	0.0063
	Residual	0.0003	1	0.0003	9.43	0.0277
	Cor Total	0.0001	5	0		
** *t* **	Model	312.06	3	104.02	58.89	0.0003
	*ω*	158.03	1	158.03	89.47	0.0002
	*δ*	118.03	1	118.03	66.82	0.0004
	*ωδ*	36	1	36	20.38	0.0063
	Residual	8.83	5	1.77		
	Cor Total	320.89	8			
** *m* **	Model	9.88	3	3.29	1132.81	<0.0001
	*ω*	6.49	1	6.49	2230.26	<0.0001
	*δ*	2.78	1	2.78	957.5	<0.0001
	*ωδ*	0.6127	1	0.6127	210.67	<0.0001
	Residual	0.0145	5	0.0029		
	Cor Total	9.9	8			

**Table 9 materials-16-04644-t009:** Comparison of predicted and measured values after parameter optimization under different support patterns.

Pattern	Category	Factors		Indicators			
		*ω* (%)	*δ* (mm)	*E* (mm)	*γ* (mm)	*t* (min)	*m* (g)
Grid	Theoretical value	7.399	1.464	0.281	1.007	7.811	18.953
	Test value	7	1.5	0.294	0.996	7.750	18.942
Concentric	Theoretical value	5.857	0.977	0.1	0.821	1.813	18.18
	Test value	6	1	0.1021	0.833	1.92	18.79
Crossover	Theoretical value	7.835	1.27	0.242	0.857	2.645	18.517
	Test value	8	1.3	0.249	0.835	2.786	18.559

## Data Availability

Not applicable.

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
