# Peer review of "Study of Biomass Composite Workpiece Support Structure Based on Selective Laser Sintering Technology"

_materials, 2023, doi:10.3390/ma16134644_

Round 1

Reviewer 1 Report

Dear Authors. The article is interesting and creates a contribution. However, some improvements should be made. Specific comments are provided in the attached file.

Author Response

Thank you for your comments, please see the document for your response.

Reviewer 2 Report

The work presents a study on selective laser sintering of thin walled structures. The authors have presented the paper nicely, however I have the following suggestions:

1. There is no point of explaining the reason for any phenomenon in abstract. The abstract must contain some significant results from the study and not just texts. 

2. The objective of the study must be clearly presented in the paper after the literature review. 

3. The exact composition of the ingredients must be presented in the manuscript with adequate evidence. 

4. Figure 3 must include description of parts in the text as well. 

5. There are several typos throughout the text, see line 270, 286, 355 etc.

6. What was the need for ANNOVA analysis in teh present context of study? Why did not the authors perform experiments as per parameter settings?

7. How many repetition was done for each set of parameters? Why did the authors refrain themselves in including error analysis?

8. It would be a good study if the authors include some soft computing methodology to carry out multi-objective optimization rather sticking to statistical traditional methods. 

9. The conclusion looks complex, it should be more concise and in point-wise manner for better visibility to the readers. 

10. Why Response surface methodology is considered in this study? I could not see and response surface graphs in the paper. 

Author Response

(The authors gave the same response as above.)

Round 2

Reviewer 1 Report

The revised version of the manuscript is acceptable.

Reviewer 2 Report

The authors have incorporated the suggested changes and the paper looks fine.